# Association between CHADS_2_, CHA_2_DS_2_-VASc, ATRIA, and Essen Stroke Risk Scores and Functional Outcomes in Acute Ischemic Stroke Patients Who Received Endovascular Thrombectomy

**DOI:** 10.3390/jcm11195599

**Published:** 2022-09-23

**Authors:** Hyung Jun Kim, Moo-Seok Park, Joonsang Yoo, Young Dae Kim, Hyungjong Park, Byung Moon Kim, Oh Young Bang, Hyeon Chang Kim, Euna Han, Dong Joon Kim, JoonNyung Heo, Jin Kyo Choi, Kyung-Yul Lee, Hye Sun Lee, Dong Hoon Shin, Hye-Yeon Choi, Sung-Il Sohn, Jeong-Ho Hong, Jong Yun Lee, Jang-Hyun Baek, Gyu Sik Kim, Woo-Keun Seo, Jong-Won Chung, Seo Hyun Kim, Sang Won Han, Joong Hyun Park, Jinkwon Kim, Yo Han Jung, Han-Jin Cho, Seong Hwan Ahn, Sung Ik Lee, Kwon-Duk Seo, Yoonkyung Chang, Hyo Suk Nam, Tae-Jin Song

**Affiliations:** 1Department of Neurology, Seoul Hospital, College of Medicine, Ewha Woman’s University, Seoul 07804, Korea; 2Department of Neurology, Yongin Severance Hospital, Yonsei University College of Medicine, Yongin 16995, Korea; 3Department of Neurology, Yonsei University College of Medicine, Seoul 03722, Korea; 4Department of Neurology, Keimyung University School of Medicine, Daegu 42601, Korea; 5Department of Radiology, Yonsei University College of Medicine, Seoul 03722, Korea; 6Department of Neurology, Samsung Medical Center, Sungkyunkwan University School of Medicine, Seoul 06351, Korea; 7Department of Preventive Medicine, Yonsei University College of Medicine, Seoul 03722, Korea; 8College of Pharmacy, Yonsei Institute for Pharmaceutical Research, Yonsei University, Incheon 21983, Korea; 9Department of Neurology, Seoul Medical Center, Seoul 02053, Korea; 10Department of Neurology, Gangnam Severance Hospital, Yonsei University College of Medicine, Seoul 06273, Korea; 11Biostatistics Collaboration Unit, Department of Research Affairs, Yonsei University College of Medicine, Seoul 03722, Korea; 12Department of Neurology, Gachon University Gil Medical Center, Incheon 21565, Korea; 13Department of Neurology, Kyung Hee University Hospital at Gangdong, Kyung Hee University School of Medicine, Seoul 05278, Korea; 14Department of Neurology, National Medical Center, Seoul 04564, Korea; 15Department of Neurology, Kangbuk Samsung Hospital, Sungkyunkwan University School of Medicine, Seoul 03181, Korea; 16Department of Neurology, National Health Insurance Service Ilsan Hospital, Goyang 10444, Korea; 17Department of Neurology, Yonsei University Wonju College of Medicine, Wonju 26426, Korea; 18Department of Neurology, Sanggye Paik Hospital, Inje University College of Medicine, Seoul 01757, Korea; 19Department of Neurology, Pusan National University School of Medicine, Busan 49241, Korea; 20Department of Neurology, Chosun University School of Medicine, Gwangju 61453, Korea; 21Department of Neurology, Sanbon Hospital, Wonkwang University School of Medicine, Gunpo 15865, Korea; 22Department of Neurology, Mokdong Hospital, College of Medicine, Ewha Woman’s University, Seoul 07985, Korea

**Keywords:** endovascular thrombectomy, functional outcome, ischemic stroke, stroke risk score

## Abstract

Background: CHADS_2_, CHA_2_DS_2_-VASc, ATRIA, and Essen stroke risk scores are used to estimate thromboembolism risk. We aimed to investigate the association between unfavorable outcomes and stroke risk scores in patients who received endovascular thrombectomy (EVT). Methods: This study was performed using data from a nationwide, multicenter registry to explore the selection criteria for patients who would benefit from reperfusion therapies. We calculated pre-admission CHADS_2_, CHA_2_DS_2_-VASc, ATRIA, and Essen scores for each patient who received EVT and compared the relationship between these scores and 3-month modified Rankin Scale (mRS) records. Results: Among the 404 patients who received EVT, 213 (52.7%) patients had unfavorable outcomes (mRS 3–6). All scores were significantly higher in patients with unfavorable outcomes than in those with favorable outcomes. Multivariable logistic regression analysis indicated that CHADS_2_ and the ATRIA score were positively correlated with unfavorable outcomes after adjusting for body mass index and variables with *p* < 0.1 in the univariable analysis (CHADS_2_ score: odds ratio [OR], 1.484; 95% confidence interval [CI], 1.290–1.950; *p* = 0.005, ATRIA score, OR, 1.128; 95% CI, 1.041–1.223; *p* = 0.004). Conclusions: The CHADS_2_ and ATRIA scores were positively correlated with unfavorable outcomes and could be used to predict unfavorable outcomes in patients who receive EVT.

## 1. Introduction

Large vessel occlusion (LVO) refers to decreased perfusion of the total or partial anterior circulation. LVO occurs due to occlusion of the internal carotid artery (ICA) or proximal middle cerebral artery (MCA, M1 segment) and has a poor prognosis [1]. The number of patients receiving endovascular thrombectomy (EVT) has increased after the recent success of EVT trials [2,3,4,5,6], and the time window for the performance of EVT has also been lengthened. Therefore, the number of patients receiving EVT is continuously increasing [5,6]. Compared with treatment with intravenous tissue plasminogen activator (tPA) alone, EVT has dramatically improved the prognoses of LVO patients [7]. Although successful recanalization predictably leads to good prognoses, this may not always be true [8]. Therefore, it is important to identify the clinical, imaging, and treatment factors correlated with functional outcomes. Factors correlated with unfavorable outcomes include old age, severe neurologic deficit, and longer time between onset and EVT [9,10,11,12]. Currently, clinical factors are not used to select EVT-eligible patients, except for age and National Institutes of Health Stroke Scale (NIHSS) scores among various factors related to unfavorable outcomes [13]. Nevertheless, we must reduce the number of patients with unfavorable outcomes despite successful recanalization after EVT. Therefore, further studies are needed to identify additional factors related to unfavorable outcomes.

Previous studies have created stroke risk scores for atrial fibrillation (AF) or ischemic stroke. The effectiveness of these scoring systems for stroke risk estimation has been demonstrated. For CHADS_2_ [14], CHA_2_DS_2_-VASc [15], and ATRIA scores [16], thromboembolic risk increases with increasing scores, mainly in AF patients. These stroke risk scores are also correlated with other vascular outcomes, such as cerebral atherosclerosis in AF patients [17]. Associations between these stroke risk scores and vascular outcomes were reported not only in AF patients but also in all stroke patients [18,19]. For the Essen score, recurrent ischemic stroke risk increases with increasing score in ischemic stroke patients [20]. The components of stroke risk scores are comorbidities and laboratory findings that can be easily obtained in the emergency room, allowing for the quick calculation of scores.

However, until recently, there have been few studies analyzing the correlation between pre-admission stroke risk scores, which can be identified in the emergency room, and unfavorable outcomes in LVO patients who received EVT. We hypothesized that the stroke risk scores would be positively correlated with unfavorable outcomes in patients receiving EVT.

## 2. Methods

### 2.1. Study Popuslation

Our study was conducted with patients included in the Selection Criteria in Endovascular Thrombectomy and Thrombolytic Therapy (SECRET) registry (Clinicaltrials.gov, NCT02964052). This registry is a national, multicenter database for acute ischemic stroke patients who received intravenous tPA and EVT [21,22]. The SECRET registry retrospectively and prospectively enrolled patients who received reperfusion therapy from January 2012 to December 2017. Patients from 15 hospitals between January 2012 and December 2017 were included retrospectively. Patients from 13 hospitals were included prospectively between November 2016 and December 2017. For prospectively enrolled patients, written informed consent was obtained from the patients themselves or their next of kin. The selection criteria and definitions of the variables included in this registry have been published [21,22]. The SECRET registry included patients who were treated according to the updated guidelines at the time of reperfusion therapy and did not establish strict exclusion criteria if the guidelines were followed. Therefore, the physicians at each stroke center decided whether to perform reperfusion treatment according to the updated guidelines. In brief, intravenous tPA was used in patients who had an ischemic stroke within 4.5 h from symptom onset and met the criteria based on guidelines with a standard dose (0.9 mg/kg) and if patients had LVO on initial angiographic studies and could be treated within 8h from symptom onset, EVT was considered [21,22]. All these patients were consecutively enrolled in the SECRET registry. All patient information was anonymized and audited by the main center. Demographic data, comorbidities, including vascular risk factors, medication history, and laboratory results, neurologic status, including severity and functional outcomes, variables of reperfusion therapy (time parameters for tPA and EVT, techniques, devices, and treatment-related complications), and imaging findings before and after reperfusion therapy were collected for all patients [19,23]. The SECRET study was conducted according to the guidelines of the Declaration of Helsinki and approved by the Institutional Review Board of the Yonsei University College of Medicine (4-2015-1196).

A total of 507 acute ischemic stroke patients in the SECRET registry who received EVT were enrolled. Our study included patients who received EVT in the anterior circulation and excluded patients without data on outcome variables (Figure 1).

Neurological status at time of admission was assessed using the NIHSS score. All accessible data related to EVT were investigated. For all LVO patients who received EVT, one of the following techniques was performed: the stent-retriever technique, a direct aspiration first pass technique (ADAPT), or combined stent-retriever and aspiration thrombectomy (Solumbra technique). The first-line technique was selected by the operator based on each patient’s clinical situation, including angiographic findings and risk factors. In some cases, if the recanalization failed or the thrombus migrated, the thrombectomy technique was changed by the operator’s choice. However, in other cases, when recanalization was successful or serious procedure-related complication occurred, the thrombectomy was not re-attempted, and the thrombectomy technique was not changed. The specific types of stent and aspiration devices used for each technique were selected by the operator. Most of the stent-retriever devices were Solitaire FR (Medtronics, Irvine, CA, USA) or Trevo (Stryker, Fremont, CA, USA) devices, and aspiration catheters were mainly Penumbra system (Penumbra, Alameda, CA, USA). The time parameters of EVT were obtained from symptom onset to success of femoral puncture (onset to puncture time) and from symptom onset to success of recanalization (onset to recanalization time) [21]. For patients whose symptom onset time was unclear, the last symptom-free time was considered the symptom onset time, and this was named the last normal time.

Imaging information was obtained through computed tomography (CT), CT angiography, magnetic resonance imaging (MRI), MR angiography, and digital subtraction angiography (DSA) performed during hospitalization [24,25,26,27]. Recanalization success was assessed using the mTICI. Unsuccessful recanalization were defined as mTICI grades of 0–2a. Functional outcomes were assessed using the mRS at 3 months. mRS information was obtained through patient interviews at outpatient clinics or via phone calls with caregivers. Unfavorable outcomes were defined as mRS grades of 3–6.

### 2.2. The Stroke Risk Scoring Systems

We investigated the variables used to calculate stroke risk scores before patient admission and calculated their scores. The variables of each stroke risk score were analyzed according to previous studies. The CHADS_2_ score assigns 1 point for age > 75, diabetes mellitus (DM), hypertension, and congestive heart failure (CHF) and 2 points for a history of stroke and transient ischemic attack (TIA) [14]. The CHA_2_DS_2_-VASc score assigns 1 point for hypertension, DM, CHF, vascular diseases, age (65–74) and female sex and 2 points for age > 75 and history of stroke and TIA [15]. The ATRIA score assigns 1 point for female sex, DM, hypertension, CHF, proteinuria, and kidney dysfunction (estimated glomerular filtration rate [eGFR] < 45 mL/min per 1.73 m^2^) and age classifications < 65, 65–74, 75–84, and ≥85 are assigned different scores according to history of stroke [16]. The Essen score assigns 1 point for hypertension, DM, history of stroke and TIA, myocardial infarction, peripheral arterial occlusive disease, other vascular diseases, and age (65–75) and 2 points for age > 75 [20].

### 2.3. Statistical Analyses

The independent *t* test or Mann–Whitney *U* test was used to compare mean values of continuous variables, and Fisher’s exact test or the chi-square test was used to compare categorical variables. Univariable and multivariable logistic regression analyses were performed to assess independent factors for unfavorable outcomes. We used two multivariable logistic analysis model. Model 1 included body mass index (BMI) and variables with *p* < 0.1 from univariable logistic regression analysis, whereas Model 2 excluded treatment factor from among the variables in Model 1. Model 2 confirmed the association between stroke risk scores and unfavorable outcomes after adjusting pre-procedural clinical factors only. Multivariable logistic regression analysis models to calculate the odd ratios (ORs), 95% confidence intervals (CIs), and *p* values. Variables commonly included in all stroke risk scores were excluded. Sensitivity analysis was performed by additionally analyzing patients with successful recanalization group and AF-related stroke group using the same statistical analyses used for all patients.

To evaluate the prediction ability of all pre-admission stroke risk scores, we conducted receiver operating characteristic (ROC) curve analysis. The area under the curve (AUC) was calculated using ROC curve analysis; the optimal cut-off values of each stroke risk score were obtained at the level with the highest Youden index (sensitivity + specificity − 1). As performance parameters for each stroke risk score, diagnostic sensitivity and specificity, positive predictive value (PPV), and negative predictive value (NPV) were analyzed using ROC curve analysis. To compare the ability of each stroke risk score to predict unfavorable outcomes, the AUC values were compared. We used the multivariable logistic regression model as the benchmark to evaluate increased associations between stroke risk scores and unfavorable outcomes in patients who received EVT. We compared AUCs to assess model discrimination and calculated category-based net reclassification improvement (NRI), continuous-based NRI, and the relative integrated discrimination improvement (IDI). All statistical analyses were performed using open-source statistical package R version 3.6.3 (R Project for Statistical Computing, Vienna, Austria), and *p* < 0.05 was considered statistically significant.

## 3. Results

### 3.1. Study Population

Of 507 patients, 6 patients without information about the modified thrombolysis in cerebral infarction (mTICI) grade immediately after EVT, 37 patients without anterior circulation occlusion, and 60 patients without 3-month-modified Rankin scale (mRS) records were excluded. In total, our study included 404 patients who received EVT (Figure 1). Finally, a total of 404 LVO patients were included in our study, and all patients received EVT. Of these patients, 191 (47.3%) had favorable outcomes (mRS 0–2), and 213 (52.7%) had unfavorable outcomes (mRS 3–6). Successful recanalization was achieved in 332 patients (82.2%). Of the successful recanalization group, 181 (54.5%) patients had favorable outcomes, and 151 (45.5%) had unfavorable outcomes. The number of patients who died within 3 months was 85 (21.0%) in the EVT group and 57 (17.2%) in the successful recanalization group (Table 1).

### 3.2. Correlation between Stroke Risk Scores and Functional Outcomes

The correlations of clinical, treatment, and imaging parameters with 3-month functional outcomes are shown in Table 1. In patients who received EVT, all stroke risk scores were significantly higher in those patients who had unfavorable outcomes than in those who had favorable outcomes (CHADS_2_ score = 2 for favorable outcomes [interquartile range, 1–2] versus 3 for unfavorable outcomes [2,3], *p <* 0.001; CHA_2_DS_2_-VASc score = 3 for favorable outcomes [2,3,4] versus 4 for unfavorable outcomes [3,4,5], *p* < 0.001; ATRIA score = 7 for favorable outcomes [2–8.5] versus 8 for unfavorable outcomes [7,8,9,10], *p* < 0.001; Essen score = 3 for favorable outcomes [2,3,4] versus 4 for unfavorable outcome [3,4], *p <* 0.001). Even in patients with successful recanalization, all stroke risk scores were significantly higher in patients who had unfavorable outcomes than for those with favorable outcomes (CHADS_2_ score = 2 for favorable outcomes [1,2] versus 2 for unfavorable outcomes [2,3], *p <* 0.001; CHA_2_DS_2_-VASc score = 3 for favorable outcomes [3,4] versus 4 for unfavorable outcomes [3,4,5], *p* < 0.001; ATRIA score = 7 for favorable outcomes [2,3,4,5,6,7,8] versus 8 for unfavorable outcomes [6,7,8,9,10], *p* < 0.001; Essen score = 3 for favorable outcomes [2,3,4] versus 4 for unfavorable outcomes [3,4], *p* = 0.026).

Univariable logistic regression analysis revealed that higher stroke risk scores were positively associated with unfavorable outcomes, along with older age, DM, eGFR < 60 mL/min, CHF, previous infarction, higher NIHSS score, aspiration alone, number of stent-retriever passes, and hemorrhagic transformation. Higher BMI, current smoking, coronary disease, combined IV/IA thrombectomy, mTICI 2b–3, and stent-retriever thrombectomy were inversely associated with unfavorable outcomes (Appendix A).

Multivariable logistic regression analysis (Model 1) revealed that CHADS_2_ and ATRIA scores were positively associated with unfavorable outcomes, and CHA2DS2-VASc scores tended to be associated with unfavorable outcomes (CHADS_2_ score = OR, 1.484 [95% CI, 1.290–1.950]; *p* = 0.005, ATRIA score = 1.128 [1.041–1.223]; *p* = 0.004) (Table 2). Model 2 revealed that all stroke risk scores were positively associated with unfavorable outcomes (CHADS_2_ score = 1.678 [1.327–2.121]; *p* < 0.001, CHA_2_DS_2_-VASc score = 1.257 [1.073–1.471], *p* = 0.005, ATRIA score = 1.091 [1.013–1.176]; *p* = 0.021, Essen score = 1.350 [1.096–1.664], *p* = 0.005) (Appendix A). Even when only patients who had successful recanalization were analyzed (Model 1), the CHADS_2_ score (1.728 [1.084–2.754); *p* = 0.022) and ATRIA scores (1.161 [1.000–1.348]; *p* = 0.049) were positively associated with unfavorable outcomes, and the CHA_2_DS_2_-VASc score tended to be associated with unfavorable outcomes (Appendix A).

In the comparison with AF-related stroke, the CHADS_2_, CHA_2_DS_2_VASc, ATRIA, and Essen scores were significantly lower in the favorable outcome group (CHADS_2_ score; median 4 [IQR 3–4] vs. 4 [IQR 3–5], *p* < 0.001) (CHA_2_DS_2_VASc score; median 2 [IQR 1–3] vs. 3 [IQR 2–3], *p <* 0.001) (ATRIA score; median 8 [IQR 6–9] vs. 9 [IQR 8–10], *p <* 0.001) (Essen score; median 3 [IQR 3–4] vs. 4 [IQR 3–4], *p =* 0.008). The above results are summarized in Appendix A. In multivariable logistic regression analysis, the CHADS_2_ score was only associated with unfavorable outcome along with BMI, eGFR < 60 mL/min, heart failure, the initial NIHSS score, combined IA/IV thrombolysis, stent-retriever alone, aspiration alone, number of stent-retriever passes, successful recanalization, and hemorrhagic transformation (*p =* 0.012) (Appendix A).

### 3.3. Comparison of Stroke Risk Scores for Unfavorable Outcomes

Of the stroke risk scores, the ATRIA score had the highest AUC value, but the difference between the ATRIA score and other scores was not significant (Table 3). In pairwise comparisons of the AUCs, there were significant differences between the AUC for the Essen score and those for the other scores (AUC of Essen score = 0.596 [95% CI, 0.542–0.650] versus AUC of CHADS_2_ score = 0.654 [0.604–0.705]; *p <* 0.001, AUC of Essen score versus AUC of CHA_2_DS_2_-VASc score = 0.644 [0.592–0.697]; *p =* 0.011, AUC of Essen score versus AUC of ATRIA score = 0.663 [0.610–0.715]; *p =* 0.014), and there were no significant differences between the AUCs of the other scores (Appendix A). Figure 2 shows the univariable and multivariable ROC curves of all stroke risk scores for unfavorable outcomes.

Additionally, when the ROC curves of patients who had successful recanalization only were analyzed, the ATRIA score also had the highest AUC value. However, the details were only significantly different between the Essen and CHADS_2_ scores (AUC of Essen score = OR, 0.570 [95% CI, 0.510–0.630] versus AUC of CHADS_2_ score = 0.621 [0.563–0.679]; *p =* 0.004) and between the Essen and ATRIA scores (AUC of Essen score versus AUC of ATRIA score = 0.642 [0.583–0.702]; *p =* 0.014) (Appendix A).

The category-based NRI was significantly increased in the model using ATRIA scores compared with the model without stroke risk scores (*p* < 0.001). The continuous-based NRI was significantly increased in the model using CHADS_2_ scores (*p* < 0.001), ATRIA scores (*p* < 0.001), and Essen scores (*p* = 0.004) compared with the model without stroke risk scores. The relative IDI was also increased in the model using ATRIA scores compared with the model without stroke risk scores (*p* = 0.001). Only the model using ATRIA scores had significantly increased category-based NRI, continuous-based NRI, and relative IDI compared with the model without stroke risk scores (Appendix A).

## 4. Discussion

The main finding of this study was a positive association between the CHADS_2_ and ATRIA scores and 3-month unfavorable outcomes in LVO patients who received EVT. Even when only patients who had undergone successful recanalization were analyzed, the CHADS_2_ and ATRIA scores were related to unfavorable outcomes.

Patients selected according to diffusion-perfusion mismatch or clinical-diffusion mismatch within a specific time period had better functional outcomes compared with those who received medical therapy only. Imaging findings, such as the volume of the infarct core and hypoperfusion tissue, and clinical findings, such as NIHSS scores, are important factors in patient selection [5,6]. However, previous studies only considered age for their inclusion criteria, and other important clinical factors were not considered when selecting EVT patients. As EVT is widely performed, studies on comorbidities that directly correlate with functional outcomes in patients receiving EVT have been conducted [28,29]. As a result, a history of DM, eGFR<60mL/min, higher glucose levels on admission, and blood pressure variability were reported to be directly related to unfavorable outcomes [28,30,31,32]. In our study, like other studies, older age, DM, and eGFR<60mL/min were correlated with unfavorable outcomes.

CHADS_2_, CHA_2_DS_2_-VASc, and ATRIA scores were created to combine the variables related to thromboembolic risk in AF patients. The Essen score was created to combine the factors related to the recurrence of ischemic stroke after an index stroke. Our study showed that the CHADS_2_ and ATRIA scores were also positively associated with unfavorable outcomes in LVO patients who received EVT. These two stroke risk scores consist of components that predict stroke severity or unfavorable outcomes. Therefore, our study suggested that not only imaging findings, but also clinical factors could be used to select EVT candidates.

In previous clinical trials, patients with successful recanalization and who received EVT were likely to have favorable outcomes [3,4,7,33,34]. Successful recanalization is more likely to influence functional outcomes than other factors that contribute to the stroke risk score. We previously reported that the CHADS_2_, CHA_2_DS_2_-VASc, ATRIA, and Essen scores could be used to predict recanalization in stroke patients receiving EVT [19]. We explored associations between stroke risk scores and functional outcomes at 3 months and found a positive association between all stroke risk scores and unfavorable outcomes after adjusting for covariates that could be identified before the procedure, whereas there was a positive association between CHADS_2_ and ATRIA scores and unfavorable outcomes in multivariable regression analysis after covariate adjustment, including successful recanalization. We then analyzed only those patients who had undergone successful recanalization; the CHADS_2_ and ATRIA scores were still positively associated with unfavorable outcomes [35]. Therefore, regardless of successful recanalization, the CHADS_2_ and ATRIA scores appear to be positively associated with unfavorable outcomes.

Compared with the other stroke risk scores, the ATRIA score is weighted for age and eGFR < 60 mL/min. The ATRIA score has been reported to be better than the CHADS_2_ or CHA_2_DS_2_-VASc scores in predicting thromboembolic risk in patients with AF [35,36]. As with previous studies, we found that the ATRIA score outperformed other stroke risk scores in predicting unfavorable outcomes in patients who received EVT. This may have been because the group who had unfavorable outcomes consisted of older patients and more patients with eGFR < 60 mL/min; these variables are weighted more in the ATRIA score. Moreover, coronary disease, one of the vascular diseases considered in calculation of the CHA_2_DS_2_-VASc and Essen scores, was more common in the group with favorable outcomes; therefore, the CHA_2_DS_2_-VASc and Essen scores did not perform well in predicting unfavorable outcomes. In addition, the Essen score was estimated to perform the worst because current smoking weighted by the Essen score did not differ between the different functional outcome groups.

Although most factors related to the occurrence of thromboembolic events in patients with AF overlap with factors related to unfavorable outcomes in patients who received EVT, other factors are not related to the unfavorable outcome or are inverse correlated with it. Therefore, a new stroke risk score that predicts unfavorable outcomes in patients who received EVT must include appropriate clinical factors. In our study, AUC for unfavorable outcomes of CHADS_2_ and ATRIA scores was 0.6–0.7 [37], which was considered as modest performance. Therefore, the inclusion of neurologic severity or treatment factor such as NIHSS or onset to visit time as well as clinical factors should be considered in the new stroke risk score to increase predictive accuracy. A new stroke risk score may be helpful in selecting patients for EVT and should be developed in the future.

### Limitations

First, although some patients included in our study were prospectively enrolled, our dataset also included retrospectively enrolled patients. In addition, patients who performed reperfusion therapy according to the updated guidelines without strict inclusion and exclusion criteria were enrolled by the decision of each stroke center. Therefore, in some cases, it is possible that patients were excluded by the decision of the physician of each stroke center, and this criterion may vary slightly from center to center, and even the number of these patients cannot be counted. Ultimately, there may have been selection bias, and we cannot conclude a causal relationship between stroke risk scores and functional outcomes. There are also 60 patients without mRS at 3 months records as missing data, which may have caused another selection bias. Second, it is difficult to extrapolate the results across all races because our registry only includes information from stroke centers in Korea. Third, CHADS_2_, CHA_2_DS_2_-VASc, and ATRIA score were developed in patients with AF, but all ischemic stroke patients were included in this study. Previous studies have already shown the potential of stroke risk score to predict vascular outcomes in patients with all ischemic stroke [18,19], and our study enrolled all ischemic stroke patients because endovascular thrombectomy is performed regardless of AF. Nevertheless, in order to provide additional information to the readers, the results of the analysis of patients with AF-related stroke are provided in the Appendix A. Fourth, the predictability of unfavorable outcomes for patients who underwent EVT using one of the most established and respected standards to estimate a 10-year stroke risk profile, such as the revised Framingham stroke risk profile, was not analyzed. This is due to the limitation of SECRET registry data that does not include EKG profile. Finally, due to the rapid development of thrombectomy techniques and devices in recent years, some results of this study may not be currently applicable.

## 5. Conclusions

Pre-admission CHADS_2_ and ATRIA scores were associated with unfavorable outcomes in LVO patients who received EVT. Therefore, these scores could predict unfavorable outcomes in LVO patients receiving EVT and even in those who underwent successful recanalization.

## Figures and Tables

**Figure 1 jcm-11-05599-f001:**
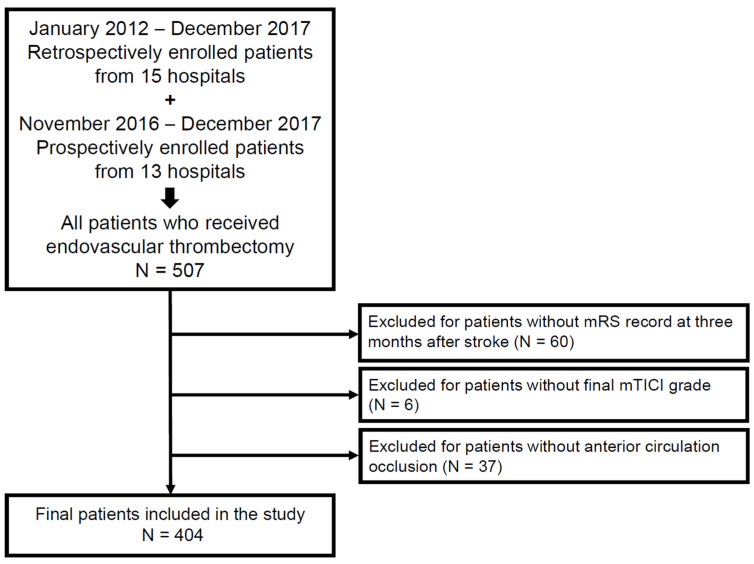
Patient selection strategy used in the study. mRS, modified Rankin Scale; mTICI, modified thrombolysis in cerebral infarction.

**Figure 2 jcm-11-05599-f002:**
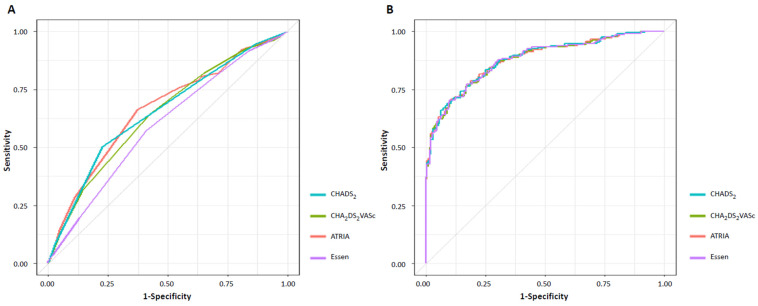
Receiver operating characteristic curve analyses of unfavorable outcomes based on stroke risk scores. (**A**) Univariable ROC analysis, (**B**) multivariable ROC analysis. ROC, receiver operating characteristic.

**Table 1 jcm-11-05599-t001:** Baseline characteristics correlated with mRS at 3 months in EVT patients and successful recanalization patients.

	EVT— Followed Up to 3 Months (N = 404)		*p*-Value	Successful Recanalization— Followed Up to 3 Months (N = 332)		*p*-Value
	Favorable Outcome (mRS 0–2) (N = 191)	Unfavorable Outcome (mRS 3–6) (N = 213)		Favorable Outcome (mRS 0–2) (N = 181)	Unfavorable Outcome (mRS 3–6) (N = 151)	
Age, years, mean (SD)	72.6 ± 13.2	79.7 ± 12.4	<0.001	73.0 ± 12.7	79.0 ± 12.3	<0.001
Female, (%)	86 (45.0%)	99 (46.5%)	0.847	83 (45.9%)	68 (45.0%)	0.969
BMI (kg/m^2^)	21.4 ± 4.0	20.2 ± 4.2	0.003	21.3 ± 3.9	20.1 ± 4.0	0.005
**Vascular risk factors**						
Hypertension, (%)	142 (74.4%)	161 (75.6%)	0.863	134 (74.0%)	107 (70.9%)	0.602
Diabetes mellitus, (%)	82 (42.9%)	148 (69.5%)	<0.001	78 (43.1%)	100 (66.2%)	<0.001
Hypercholesterolemia, (%)	85 (44.5%)	96 (45.1%)	0.989	81 (44.8%)	68 (45.0%)	>0.999
Current smoking, (%)	41 (21.5%)	29 (13.6%)	0.051	36 (19.9%)	19 (12.6%)	0.102
eGFR < 60 mL/min, (%)	72 (37.7%)	131 (61.5%)	<0.001	67 (37.0%)	93 (61.6%)	<0.001
**Comorbidities**						
Atrial fibrillation (%)	95 (49.7%)	126 (59.2%)	0.072	90 (49.7%)	90 (59.6%)	0.091
Heart failure, (%)	11 (5.8%)	26 (12.2%)	0.038	11 (6.1%)	17 (11.3%)	0.135
Coronary disease, (%)	67 (35.1%)	53 (24.9%)	0.033	66 (36.5%)	40 (26.5%)	0.068
Peripheral artery disease, (%)	5 (2.6%)	10 (4.7%)	0.402	4 (2.2%)	6 (4.0%)	0.539
Previous infarction, (%)	34 (17.8%)	57 (26.8%)	0.042	32 (17.7%)	39 (25.8%)	0.095
Previous hemorrhage	7 (3.7%)	13 (6.1%)	0.369	7 (3.9%)	8 (5.3%)	0.719
**Medication before admission**						
Prior antiplatelet therapy, (%)	57 (29.8%)	74 (34.7%)	0.345	55 (30.4%)	52 (34.4%)	0.504
Prior anticoagulation therapy, (%)	38 (19.9%)	29 (13.6%)	0.119	37 (20.4%)	21 (13.9%)	0.157
Prior statin therapy, (%)	58 (30.4%)	68 (31.9%)	0.818	56 (30.9%)	51 (33.8%)	0.665
Initial NIHSS score, median ((IQR)	12 (7–16)	18 (13–21)	<0.001	12 (7–16)	17 (13–20.5)	<0.001
**Treatment**						
IA thrombectomy alone, (%)	100 (52.4%)	145 (68.1%)	0.002	95 (52.5%)	99 (65.6%)	0.022
Combined IV/IA thrombolysis *, (%)	91 (47.6%)	68 (31.9%)	0.002	86 (47.5%)	52 (34.4%)	0.022
Stent-retriever alone, (%)	154 (80.6%)	144 (67.6%)	0.004	176 (97.2%)	141 (93.4%)	0.155
Aspiration alone, (%)	5 (2.6%)	18 (8.5%)	0.021	5 (2.76%)	10 (6.62%)	0.1553
Combined stent-retriever/aspiration **, (%)	32 (16.8%)	51 (23.9%)	0.096	31 (17.1%)	28 (18.5%)	0.848
Number of stent-retrieval passes (SD)	1.8 ± 1.3	2.4 ± 2.3	<0.001	1.7 ± 1.2	2.3 ± 2.0	0.004
Onset to puncture, min, mean (SD)	343.1 ± 447.6	335.4 ± 301.0	0.842	344.1 ± 457.3	324.9 ± 299.1	0.646
Onset to recanalization, min, mean (SD)	N/A	N/A		418.5 ± 482.8	408.7 ± 303.9	0.825
LNT-to-puncture time (within 6 h)	142 (74.4%)	150 (70.4%)	0.442	134 (74.0%)	106 (70.2%)	0.513
**Stroke etiology**			0.418			0.258
Cardioembolic	103 (53.9%)	124 (58.2%)		98 (54.1%)	89 (58.9%)	
Large artery atherosclerosis	35 (18.3%)	29 (13.6%)		32 (17.7%)	17 (11.3%)	
Undetermined or others	53 (27.8%)	60 (28.2%)		51 (28.2%)	45 (29.8%)	
**Image finding after EVT**						
mTICI 2b-3	181 (94.8%)	151 (70.9%)	<0.001	N/A	N/A	N/A
Hemorrhagic transformation	9 (4.7%)	65 (30.6%)	<0.001	54 (29.8%)	87 (57.6%)	<0.001
**Pre-admission stroke risk score, median (IQR)**						
CHADS_2_ score	2 (1–2)	3 (2–3)	<0.001	2 (1–2)	2 (2–3)	<0.001
CHA_2_DS_2_VASc score	3 (2–4)	4 (3–5)	<0.001	3 (2–4)	4 (3–5)	<0.001
ATRIA score	7 (2–8.5)	8 (7–10)	<0.001	7 (2–8)	8 (6–10)	<0.001
Essen score	3 (2–4)	4 (3–4)	<0.001	3 (2–4)	4 (3–4)	0.026

mRS, modified Rankin Scale; EVT, endovascular thrombectomy; SD, standard deviation; BMI, body mass index; eGFR, estimated glomerular filtration rate; National Institutes of Health Stroke Scale, NIHSS; IQR, interquartile range; IA; IV, intravenous; intra-arterial; IV, LNT, last normal time. * administration of tissue plasminogen activator prior to endovascular thrombectomy. ** cases in which stent retriever and aspiration were performed simultaneously or sequentially.

**Table 2 jcm-11-05599-t002:** Multivariate analysis for stroke risk score correlated with the unfavorable outcome among 404 patients with EVT (Model 1).

	CHADS_2_	*p*-Value	CHA_2_DS_2_VASc	*p*-Value	ATRIA	*p*-Value	Essen	*p*-Value
Variables	OR (95% CI)		OR (95% CI)		OR (95% CI)		OR (95% CI)	
BMI, per-1-kg/m^2^ increase	0.940 (0.878–1.006)	0.075	0.966 (0.899–1.038)	0.075	0.966 (0.899–1.038)	0.341	0.962 (0.985–1.034)	0.297
Current smoking	0.545 (0.253–1.173)	0.121	0.533 (0.248–1.147)	0.108	0.533 (0.247–1.148)	0.108	0.466 (0.220–0.989)	0.047
eGFR < 60 mL/min	1.484 (0.812–2.714)	0.200	1.691 (0.961–2.973)	0.068	1.542 (0.802–2.967)	0.194	1.897 (1.108–3.246)	0.020
Atrial fibrillation	0.913 (0.515–1.619)	0.755	0.949 (0.536–1.678)	0.856	0.955 (0.542–1.684)	0.874	0.931 (0.526–1.649)	0.806
Heart failure	1.418 (0.484–4.158)	0.525	1.671 (0.550–5.073)	0.365	1.918 (0.706–5.210)	0.202	2.265 (0.895–5.733)	0.845
Coronary disease	0.479 (0.267–0.858)	0.013	0.498 (0.278–0.889)	0.018	0.506 (0.285–0.897)	0.020	0.511 (0.287–0.910)	0.022
Previous infarction	1.675 (0.868–3.233)	0.124	1.649 (0.874–3.111)	0.123	1.427 (0.655–3.108)	0.371	1.801 (0.954–3.426)	0.070
Initial NIHSS score, per 1-score increase	1.183 (1.126–1.243)	<0.001	1.176 (1.120–1.236)	<0.001	1.178 (1.121–1.238)	<0.001	1.180 (1.123–1.239)	<0.001
**IV thrombolysis**								
IA thrombolysis alone	Reference		Reference		Reference		Reference	
Combined IA/IV thrombolysis *	0.420 (0.243–0.727)	0.002	0.429 (0.248–0.744)	0.003	0.428 (0.248–0.738)	0.002	0.448 (0.257–0.781	0.005
**IA thrombolysis**								
Stent-retriever alone	0.961 (0.465–1.984)	0.914	0.922 (0.447–1.904)	0.827	0.915 (0.446–1.877)	0.808	0.938 (0.455–1.935)	0.863
Aspiration alone	7.361 (1.978–27.386)	0.003	7.700 (2.033–29.158)	0.003	7.796 (2.068–29.390)	0.002	8.128 (2.112–31.285)	0.002
Number of stent-retriever passes, per-1-passes increase	1.169 (0.986–1.384)	0.072	1.178 (0.996–1.393)	0.056	1.179 (0.998–1.393)	0.053	1.180 (0.999–1.393)	0.052
**Imaging finding after EVT**								
mTICI 2b-3	0.142 (0.059–0.340)	<0.001	0.138 (0.058–0.330)	<0.001	0.140 (0.059–0.334)	<0.001	0.131 (0.055–0.312)	<0.001
Hemorrhagic transformation	11.314 (4.836–26.468)	<0.001	12.450 (5.274–29.391)	<0.001	13.394 (5.681–31.589)	<0.001	13.304 (5.635–31.413)	<0.001
**Risk scoring score**								
Per-1-point increase	1.484 (1.290–1.950)	0.005	1.177 (0.978–1.416)	0.085	1.128 (1.041–1.223)	0.004	1.173 (0.903–1.524)	0.231

EVT, endovascular thrombectomy; OR, odd ratio; CI, confidence interval; BMI, body mass index; eGFR, estimated glomerular filtration rate; National Institutes of Health Stroke Scale, NIHSS; IV, intravenous; IA, intra-arterial; mTICI, modified thrombolysis in cerebral infarction. * administration of tissue plasminogen activator prior to endovascular thrombectomy.

**Table 3 jcm-11-05599-t003:** Receiver operating characteristic curve analysis of risk scores for the probability of a functional outcomes.

	AUC	Optimal Cutoff	Diagnostic Sensitivity	Diagnostic Specificity	PPV	NPV
**Unfavorable outcome: EVT patients**						
Pre-admission CHA_2_DS_2_VASc	0.644	3.5	0.643	0.576	0.628	0.591
Pre-admission CHADS2	0.654	2.5	0.502	0.775	0.713	0.583
Pre-admission ATRIA	0.663	7.5	0.662	0.628	0.665	0.625
Pre-admission Essen	0.596	3.5	0.573	0.592	0.61	0.554
**Unfavorable outcome: successful recanalization patients**						
Pre-admission CHA_2_DS_2_VASc	0.613	3.5	0.603	0.564	0.535	0.63
Pre-admission CHADS_2_	0.621	2.5	0.431	0.774	0.613	0.62
Pre-admission ATRIA	0.642	7.5	0.636	0.63	0.589	0.675
Pre-admission Essen	0.57	3.5	0.523	0.586	0.513	0.596

AUC, area under curve; PPV, positive predictive value; NPV, negative predictive value; EVT, endovascular thrombectomy.

## Data Availability

The data presented in this study are available on request from the corresponding authors. The data are not publicly available due to privacy.

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
