# Peer review of "Association between CHADS2, CHA2DS2-VASc, ATRIA, and Essen Stroke Risk Scores and Functional Outcomes in Acute Ischemic Stroke Patients Who Received Endovascular Thrombectomy"

_jcm, 2022, doi:10.3390/jcm11195599_

Round 1

Reviewer 1 Report

Dear Sir/Madam,

I had the opportunity to act as a reviewer on the recent submission by Kim et al. to the Journal of Clinical Medicine.

The authors present original research investigates the association between unfavorable outcomes and stroke risk scores in patients who received endovascular thrombectomy. They found that the CHADS2 and ATRIA scores positively correlate with unfavorable outcomes and could be used to predict unfavorable outcomes in patients who receive endovascular thrombectomy.

The manuscript is well structured; however, some issues need to be addressed:

  1. Please comment on choosing and studying stroke risk scores validated for patients with atrial fibrillation in a mixed cohort of patients with and without atrial fibrillation. It would be more interesting to know how a stroke risk score such as the revised Framingham stroke risk profile performs.
  2. The area under the curve (AUC) showed a rather modest performance (under 0.7) of these scores. How do the authors comment that and what could be added to these scores in order to improve the AUC?

Best regards,

Author Response

Dear reviewers & editor

First of all, we would like to thank the reviewers and editorial board members for their time and valuable comments on our manuscript. We have addressed our opinions on each comment from two reviewers in this response letter, and made several changes and corrections to our original manuscript.

We tried to revise our manuscript according to reviewer’s suggestions as much as possible, and revised parts are written in red color texts in the manuscript. We hope that the revisions in the manuscript and our accompanying responses will be sufficient to make our manuscript suitable for publication in Journal of Clinical Medicine.

We shall look forward to hearing from you at your earliest convenience.

Sincerely yours,

Tae-Jin Song, MD, PhD.

Department of Neurology, Seoul Hospital, Ewha Womans University College of Medicine, 260, Gonghang-daero, Gangseo-gu, Seoul, 07804, Republic of Korea

Tel: +82-2-2650-2677, Fax: +82-2-2650-5958; E-mail: [email protected]

Hyo Suk Nam, MD, PhD

Department of Neurology, Yonsei University College of Medicine

50-1 Yonsei-ro, Seodaemoon-gu, Seoul, 03722, Korea

Tel: +82-2-2228-1617, Fax: 82-2-393-0705; E-mail: [email protected]

Reviewer 2 Report

Kim et al. aimed to investigate the association between unfavorable outcomes and stroke risk scores in patients who received endovascular thrombectomy (EVT). They found that CHADS2 and ATRIA scores were positively correlated with unfavorable outcomes. They conclude that theses scores could be used to predict unfavorable outcomes in patients who received EVT.

Method

-Could you highlight more the inclusion and exclusion criteria?

-Page 3: “Therefore 6 patients..” -> “in total…”. This paragraph should be in the result section in my opinion.

Results

-I wonder why you do not try to build your own scoring system? You have sufficient data to do so.

-I would not have included procedural data in the multivariate analysis since these data are not available when you have to decided if the patient should or should not underwent EVT. Are the results different if you do it this way?

-You might not put all the ORs that are already in the tables, to facilitate reading.

Discussion

-You should add in the limitations section that there is a another recruitment bias. Indeed, some patients were excluded from the EVT (“the physicians at each stroke center decided whether to perform reperfusion treatment according to the updated guidelines”). Moreover, we don’t know the number of patients excluded for this reason.

-The 60 patients excluded for patients without mRS record at three months after stroke add another bias that you should discuss in the limitation section. Are they alive?

Author Response

(The authors gave the same response as above.)

Round 2

Reviewer 1 Report

Dear Sir/Madam,

Thank you for reviewing the manuscript and addressing the mentioned issues. These were adequately answered. Therefore, the manuscript seems suitable for publishing in the present form.

Best regards

Reviewer 2 Report

The manuscript has been sufficently improved to warrant publication.